



# High-spatiotemporal reconstruction of biogeochemical dynamics in Australia integrating satellites products and in-situ observations (2000–2022)

Xiaohan Zhang[1], Lizhe Wang[1,*], Jining Yan[1], and Sheng Wang[1]

[1]School of Computer Science, China University of Geosciences, Wuhan, 430074, China

**Correspondence:** Lizhe Wang (lizhe.wang@gmail.com)

**Abstract.** The marine biogeochemical time-series products, which include total alkalinity, inorganic carbon, nitrate, phosphate, silicate, and pH, constitute a foundational support mechanism for the ongoing surveillance of oceanic biogeochemical changes. These products play a critical role in facilitating research focused on dynamic monitoring of marine ecosystems and fostering sustainable oceanic development. However, existing monitoring methodologies are hampered by inherent limitations, notably

the paucity of observational products that simultaneously offer high spatial and temporal resolutions. Furthermore, the inter-polation methods typically employed in these contexts frequently prove low-effective on a large scale, resulting in data with extensive temporal and spatial expanses that are difficulty for applications aimed at monitoring large-scale ocean dynamics. A novel integration of the CANYON-B and Random Forest regression methods was explored to address these challenges in reconstructing key marine biogeochemical parameters. This work reconstructs the concentrations of these marine biogeochem-

icals at the sea surface within Australia's Exclusive Economic Zone over the period from 2000 to 2022 on a 1-kilometre scale. The approach involves the amalgamation of multi-source in-situ ocean chemistry time-series observations with MODIS Terra ocean reflectance imagery and ocean water colour product distributions. This research highlights the substantial capabilities of machine learning for the large-scale reconstruction of ocean chemistry data, introducing a new, viable method for utilising in-situ measurements and optical imagery in reconstructing marine biogeochemical elements, thereby significantly enhancing

our ability to monitor large-scale ocean dynamics. The datasets generated and analysed in this study are available on Science Data Bank (https://doi.org/10.57760/sciencedb.09331)(Zhang et al., 2024).

## 1 Introduction

Monitoring the biogeochemical properties of the ocean, including total alkalinity ($A_T$), inorganic carbon ($C_T$), nitrate($NO_3^-$), phosphate($PO_4^{3-}$), silicate($Si(OH)_4$), and pH, is of paramount importance for evaluating the health of marine ecosystems,

deciphering biogeochemical cycles, and forecasting the effects of climate change. These parameters serve as essential indicators of the ocean's capacity to buffer acidifying inputs, the nutritional status affecting primary productivity, and overall water quality (Chai et al., 2020). Through regular and systematic monitoring of ocean chemistry parameters, researchers are able to track key variables such as the level of acidification of seawater, nutrient concentrations and the distribution of harmful pollutants (Walters, 1997). These data are not only essential for studying the health and biodiversity of marine ecosystems, but also





play an indispensable role inassessing the resilience of oceanic environments to anthropogenic stresses, predicting the impacts of global climate change and the ocean future state, developing effective sustainable fisheries policies and protecting marine life. However, traditional methods for monitoring ocean chemistry face significant challenges, particularly in terms of spatial and temporal resolution (Palmer et al., 2015; Kavanaugh et al., 2016; Palmer et al., 2015). This often results in datasets that are insufficient for analyzing complex marine processes on a large scale, thereby hindering effective policy-making and environmental management.

In the pursuit of understanding the vast and complex dynamics of the world's oceans, large-scale observational datasets of ocean chemistry have become indispensable tools (Dickey et al., 2006; Levin et al., 2019; Miloslavich et al., 2018). These datasets, which track variables such as temperature, salinity, dissolved gases, and nutrient concentrations across global scales, are crucial for studying marine ecosystems, assessing environmental health, and forecasting changes related to climate dynamics. Notably, projects like the Global Ocean Data Analysis Project (GLODAP)(Key et al., 2004, 2015; Olsen et al., 2016; Lauvset et al., 2024) and the World Ocean Atlas (Levitus et al., 1994; Antonov and Levitus, 2006; Zweng et al., 2019) have provided extensive baselines of marine biogeochemical properties, facilitating countless studies in oceanography and related disciplines.

However, despite these advancements, the field faces significant challenges, particularly regarding the spatial and temporal resolution of data (Han et al., 2023; Zhang et al., 2023). Most large-scale ocean chemistry datasets are derived from infrequent ship-based surveys or fixed-point observatories, which are then interpolated to create continuous spatial fields. This interpolation, while necessary, introduces substantial uncertainties, particularly in dynamic regions where biogeochemical properties can vary significantly over short distances and time periods. Traditional interpolation methods, such as inverse distance weighting or kriging, often assume spatial homogeneity and may not adequately capture complex gradients or the temporal variability of ocean processes (Li and Heap, 2014; Ly et al., 2013). Such shortcomings can lead to misleading representations of marine biogeochemical environments, potentially skewing our understanding of oceanic processes and their responses to environmental changes.

Moreover, these traditional methods struggle with data-sparse regions, which are plentiful in the ocean due to the logistical challenges and high costs associated with data collection. In response to these challenges, there is a growing shift towards integrating more advanced statistical and machine learning techniques, which promise to improve the resolution and accuracy of interpolated ocean chemistry data (Sagan et al., 2020; Sun and Scanlon, 2019). Recent advances in remote sensing and machine learning offer promising solutions to these challenges. This study leverages cutting-edge machine learning techniques, specifically the integration of CANYON-B models and random forest regression, to enhance the reconstruction of marine biogeochemical data across extensive spatial and temporal scales. By combining multi-source in-situ measurements with satellite imagery, this approach seeks to improve the temporal and spatial resolution of marine biogeochemical datasets, providing a more robust framework for monitoring and understanding the vast and dynamic marine environments within Australia's EEZ. This paper outlines the methodologies employed, discusses the integration of diverse data streams, and demonstrates the potential of these technologies to transform our understanding of ocean chemistry on a global scale.





## 2 Study Area and Input Data

### 2.1 Study area

The marine area of interest in this study falls within Australia's Exclusive Economic Zone (EEZ). The extent of the EEZ is shown in Figure 1. The marine area extending 200 nautical miles (about 370 km) from Australia's coastline covers an area of more than 8.8 million square kilometres and is rich in marine resources, including aquatic biological resources, oil, and natural gas. In addition, the Australian Government attaches great importance to the protection of marine ecosystems while exploiting marine resources, and many areas within the EEZ have been designated as Marine Protected Areas to protect biodiversity and maintain the balance of marine ecosystems. Marine monitoring data are pivotal, providing ongoing surveillance and analytical capabilities essential for the efficacious management of resources, and the monitoring and protection of the marine environment within the Australian EEZ.

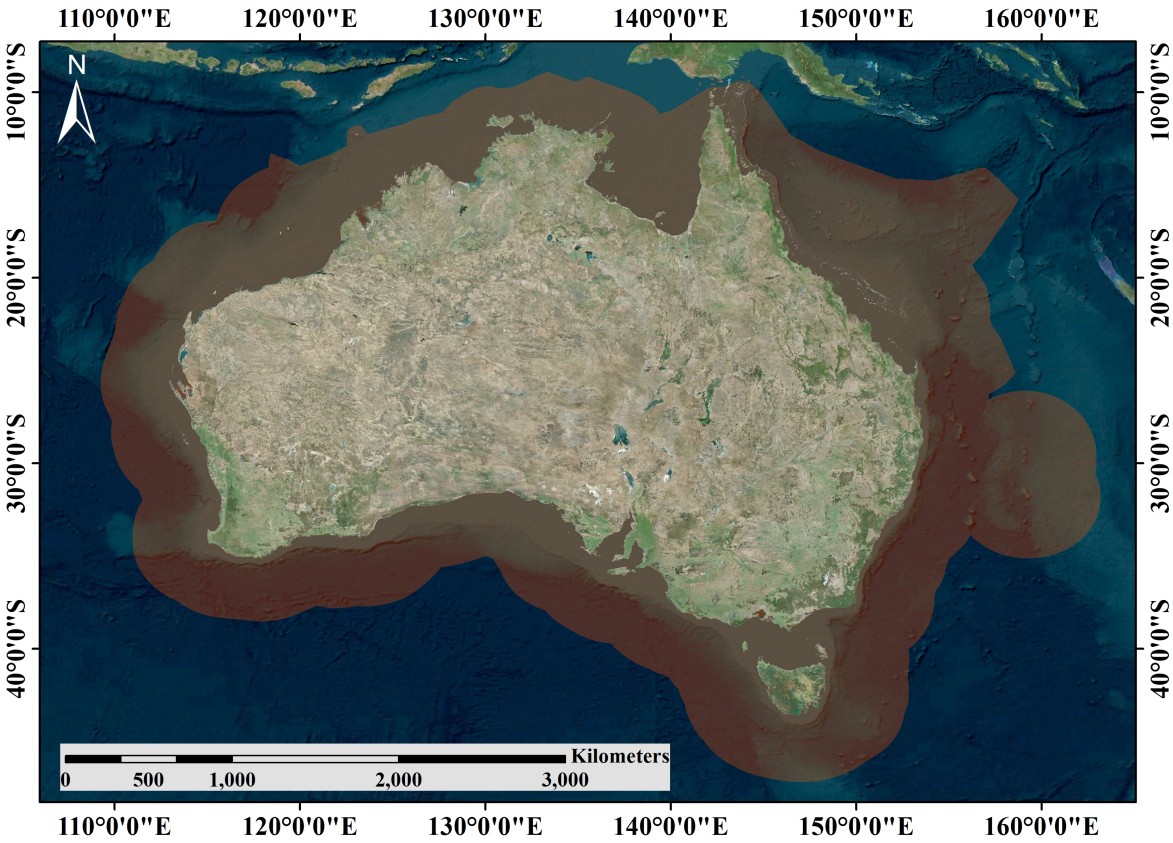

**Figure 1.** Maritime EEZ of Australia. The extent obtained from Geoscience Australia of Australian Government. Satellite image obtained from Bing Maps © Microsoft.

## 2.2 In-situ Ocean Observations Data

**Figure 2.** Distribution of in situ ocean observing data sets in the AEEZ. Satellite image obtained from Bing Maps © Microsoft.

The ocean observations employed for model construction in this study encompassed three categories of data: Argo buoys (Wong et al., 2020; Notarstefano et al., 2020), glider observations and Australian National Mooring Network Facility (ANMN) time-series observations. The distribution of data for all utilisations is presented in Figure 2.

Glider observations (Green dots in Figure 2) were collected by a fleet of eight gliders deployed around Australia by the Australian National Facility for Ocean Gliders (ANFOG), with funding from the Integrated Marine Observing System (IMOS)

and the National Collaborative Research Infrastructure Strategy (NCRIS). The data presented in the record is represented by delayed mode point data. The sensors are capable of recording temperature, salinity, dissolved oxygen, turbidity, dissolved organic matter, and chlorophyll elements based on location and depth.





ANMN (Red dots in Figure 2) is a series of national reference stations and regional moorings designed to monitor specific oceanographic phenomena in Australia's coastal waters. Water Quality Monitors (WQMs) collect a time series of physical and
'biogeochemical' data, including temperature, pressure, salinity, conductivity, depth, dissolved oxygen, chlorophyll, turbidity, and fluorescence. The WQMs and a number of NXIC CTDs located at one or more depths collect the data, which is then cleaned and averaged to produce products. A series of National Reference Stations (NRS) and several regional stations in the Queensland, Northern Australia, and New South Wales ANMN sub-facilities provide data for the production of averaged data products.

The Argo programme is a global network of ocean observations (Blue dots in Figure 2). Argo Australia is part of the Argo programme, which is led by Australian science agencies and, in particular, implemented in collaboration with Australia's marine and atmospheric science agencies. The programme is designed to collect data on the ocean's temperature, salinity, and current velocities through a series of automated submerged buoys called Argo buoys. These buoys are equipped with a series of automated submersible beacons (known as Argo buoys). The buoys are capable of automatically diving to a depth of
approximately 2000 metres and then surfacing to the surface, where they continuously measure and record data, which is then sent back to a ground station via a satellite system.

The three datasets were obtained from the Australian Ocean Data Network (AODN, https://portal.aodn.org.au/search), which records a long-term series of changes in temperature, salinity, dissolved oxygen, and other elements at specific locations in the ocean of Australia's EEZ. In accordance with the data quality levels provided in the files, low-quality, invalid, and erroneous data were cleansed to ensure the reliability of the input data.

**Table 1.** Multi-source input datasets

| Dataset Name | Data Type | Contains Elements | Timespan | Source |
|---|---|---|---|---|
| MODIS Terra Marine Reflectance Products | Raster | 1km ocean reflectance data from Terra MODIS bands 8-16 | 2000–2022 | GEE |
| MODIS Terra Ocean Water Color Products | Raster | Chl-a-POC-SST(Resampling from 4km to 1km spatial resolution) | 2000–2022 | GEE |
| Australian Glider Observations | Point | Chl-a-Current-Optical properties-Oxygen-Salinity-Temperature-Water pressure | 2008–2024 | AODN |
| ANMN Monitor Data | Point | Chl-a-Oxygen-Salinity-Temperature-Turbidity-Water pressure | 2007–2024 | AODN |
| Argo Buoy Data | Point | Salinity-Temperature-Water pressure | 1999–2024 | AODN |


## 2.3 MODIS Imagery and Products

The MODIS on the Terra satellite represents a pivotal instrument in Earth observation technology, launched by NASA in 1999 as part of the Earth Observing System (EOS) (Running et al., 1994; Guenther et al., 2002). Terra, the flagship of the EOS, carries MODIS as a key sensor designed to provide data in a wide spectral range for the purpose of Earth's surface and
atmospheric mapping at varying spatial resolutions. MODIS encompasses 36 spectral bands ranging from visible to thermal infrared wavelengths, allowing for comprehensive global observations of terrestrial, oceanic, and atmospheric variables (Justice et al., 1998). Its daily global coverage provides invaluable data for understanding Earth's biodiversity, water resources, and the impact of climate change, underscoring its role in global environmental research and monitoring. The longevity and continuity



of the MODIS data record make it exceptionally valuable for the scientific community, aiding in long-term climate studies and
environmental monitoring, thus contributing profoundly to our understanding of the Earth system.

In the scope of this research, the MODOCGA Version 6 Level 2 Gridded Lite (L2G-lite) Ocean Reflectance imagery, retrieved from Google Earth Engine (GEE) with a spatial resolution of 1 km, served as a pivotal data source(Tamiminia et al., 2020). This dataset comprises ocean surface spectral reflectance estimates derived from the Terra Moderate Resolution Imaging Spectroradiometer (MODIS) across bands 8 to 16 (Vermote and Wolfe, 2015). These estimates have been rigorously corrected for atmospheric disturbances including gaseous emissions, aerosols, and Rayleigh scattering (Feng et al., 2013). To enhance the data's reliability and minimize discrepancies caused by outliers, such as clouds and cloud shadows, a methodology involving the $median()$ function in GEE was employed to aggregate images on a monthly basis, resulting in the creation of 265 high-quality synthetic images spanning from February 2000 to February 2022.

Furthermore, MODIS water color data, also accessed via GEE, comprises the Standard Mapped Image MODIS Terra Ocean Color Data produced by the Earth Observing System Data and Information System (EOSDIS). This dataset, which has a spatial resolution of 4.6 km, includes 14 distinct spectral bands that encompass both water color products and remotely sensed reflectance data (Kilpatrick et al., 2015). To maintain consistency in the spatial and temporal resolution with the MODIS ocean reflectance imagery, several synthetic products—including monthly chlorophyll-a concentrations (Chl-a), standardized fluorescence line height, particulate organic carbon (POC), and sea surface temperature (SST) from 2000-2022—were processed. This processing involved using the median of all images within a month, followed by interpolation to a spatial resolution of 1 km, thereby ensuring that the products, spatial scales, and temporal scales were aligned across the datasets.

## 3 Method

This research comprises two principal components. Firstly, we employ the CANYON-B model to accurately calculate key ocean chemistry parameters, including $A_T$, $C_T$, and pH levels. CANYON-B utilises a Bayesian neural network approach, integrating various physical and geographical data inputs to yield high-precision biogeochemical estimations of the marine environment.

Secondly, robust correlations are established between these biogeochemical parameters and satellite-derived MODIS products, which contain ocean reflectance images and water color products. This is achieved through the application of random forest regression, a powerful machine learning technique known for its efficacy in handling complex, nonlinear data relationships. This method facilitates the integration of remotely sensed data with in-situ biogeochemical measurements, with the aim of improving the predictive accuracy of oceanic properties across large spatial scales.

The specific workflow of our methodology is meticulously illustrated in Figure 3, which details each step from data acquisition through model application to final analysis.

Earth System
Open Access   Science
Data   Discussions

**Figure 3.** Process of long-term series Australian marine biogeochemical distribution reconstruction.



## 3.1 CANYON-B

The CANYON-B model, an enhancement of the traditional CANYON framework (Sauzède et al., 2017), incorporates Bayesian neural network methodologies to predict essential biogeochemical variables in marine environments, specifically $A_T$, $C_T$, and pH. These variables are crucial for understanding oceanic carbon systems and are deduced from a range of physical oceanographic parameters, including temperature, salinity, dissolved oxygen, water pressure, and geolocation data (Bittig et al., 2018). By integrating Bayesian methods, CANYON-B significantly refines the accuracy of predictions and enriches the model's

capability to quantify uncertainties associated with these predictions. The model has undergone rigorous validation across multiple datasets, including the independent GO-SHIP bottle and sensor data. Validation results consistently demonstrate that CANYON-B provides highly accurate predictions with minimized uncertainties, affirming its effectiveness in modeling complex oceanic biogeochemical processes.

## 3.2 Random Forest Regression

Random forest regression is an integrated learning-based regression algorithm that makes regression predictions by constructing multiple decision trees and combining their predictions. Random forests can handle high-dimensional data and are robust to outliers and noise (Segal, 2004). The predictive output of a random forest regression model can usually be expressed by the following Eq.1.

$$\hat{y} = \frac{1}{N} \sum_{i=1}^{N} T_i(x) \tag{1}$$

where $\hat{y}$ is the prediction of Random Forest Regression, $N$ is the number of the number of decision trees in the Random Forest Regression, $T_i(x)$ is the prediction of the $ith$ tree for the input feature $x$. The output of a random forest regression model is the average of the predicted values of all its decision trees for a particular input $x$. Each tree independently makes a prediction for the same data point, and then all these predictions are averaged or aggregated to get the final prediction. This approach helps to reduce the variance of the model and improves the accuracy and stability of the predictions.

## 155   3.3 Reconstructing Distributional Verification

This research employed cross-validation, a widely recognized technique in machine learning, to substantiate the experimental framework. The observational dataset was partitioned into a training set, comprising 80% of the data, and a test set, comprising the remaining 20%. The training set was utilized for the calibration of the machine learning model and the reconstruction of the results, whereas the test set served as independent data for the validation of the model's performance. To ensure the development

of robust and efficacious model experiments, the 5-fold cross-validation method was implemented (Fushiki, 2011). Herein, all in situ data were methodically segregated into five subsets based on the year of acquisition and the geographic distribution of the locations, guaranteeing a uniform distribution across the observation period and the targeted area. In each iteration of the validation process, four subsets were designated as training data and one subset as test data. This procedure was repeated five times to ensure that each subset was employed in both the training and validation phases. The model's accuracy was



quantitatively assessed by calculating the mean value of the discrepancies between the validation data and the model-generated
        data across the five experiments.

        In the context of evaluating the performance of predictive models in environmental science, two statistical metrics are
        commonly utilized: the Root Mean Square Error (RMSE) and the Coefficient of Determination ($R^2$) (Chicco et al., 2021).
        RMSE is a widely used measure of the difference between values predicted by a model and the values actually observed. The

formula for RMSE is given by Eq.2. $R^2$, also known as R-squared, quantifies the proportion of the variance in the dependent
        variable that is predictable from the independent variables. This metric provides a sense of how well unseen samples are likely
        to be predicted by the model, relative to the mean of the observed data. The formula for $R^2$ presented in Eq.3.

$$\text{RMSE} = \sqrt{\frac{1}{n}\sum_{i=1}^{n}(P_i - O_i)^2} \tag{2}$$

$$R^2 = 1 - \frac{\sum_{i=1}^{n}(O_i - P_i)^2}{\sum_{i=1}^{n}(O_i - \overline{O})^2} \tag{3}$$

        where $P_i$ represents the predicted values, $O_i$ represents the observed values, is the number of observations, and $\overline{O}$ is the mean
        of the observed values.

### 3.4   Data Pre-processing

        In the course of our study, concentrations of dissolved substances were initially recorded in micromoles per litre ($\mu$mol/L). To

facilitate comparisons with measurements taken under different conditions and to comply with standard oceanographic prac-
        tices, these concentrations were converted to micromoles per kilogram ($\mu$mol/kg) using the Gibbs Seawater (GSW) Oceano-
        graphic Toolbox (McDougall and Barker, 2011). This conversion is imperative as it accounts for variations in seawater density
        influenced by temperature, salinity, and pressure, parameters that can significantly alter the mass of seawater per unit volume.
        Utilizing the precise coordinates of the National Reference Stations, the temperature, salinity, and pressure data were input

into the GSW Toolbox to compute the accurate seawater density at each location. The toolbox applies the internationally rec-
        ognized Thermodynamic Equation of Seawater – 2010 (TEOS-10) standards, ensuring that our density calculations reflect the
        state-of-the-art understanding of seawater properties (Stewart et al., 2017). These density values were then used to convert the
        original concentration measurements from $\mu$mol/L to $\mu$mol/kg, thus normalizing the data for differences in seawater matrix
        arising from natural environmental variations. This methodological approach enhances the robustness and comparability of our

chemical data across diverse marine environments.

## 4   Result and Discussion

### 4.1   Monthly Trend of Reconstruction Products

        Figure 4 provides a comprehensive visualisation of long-term trends in $A_T$ across the Australian EEZ over a twenty-year period,
        presented on a monthly basis from January to December. January to March: These months show increasing $A_T$ concentrations

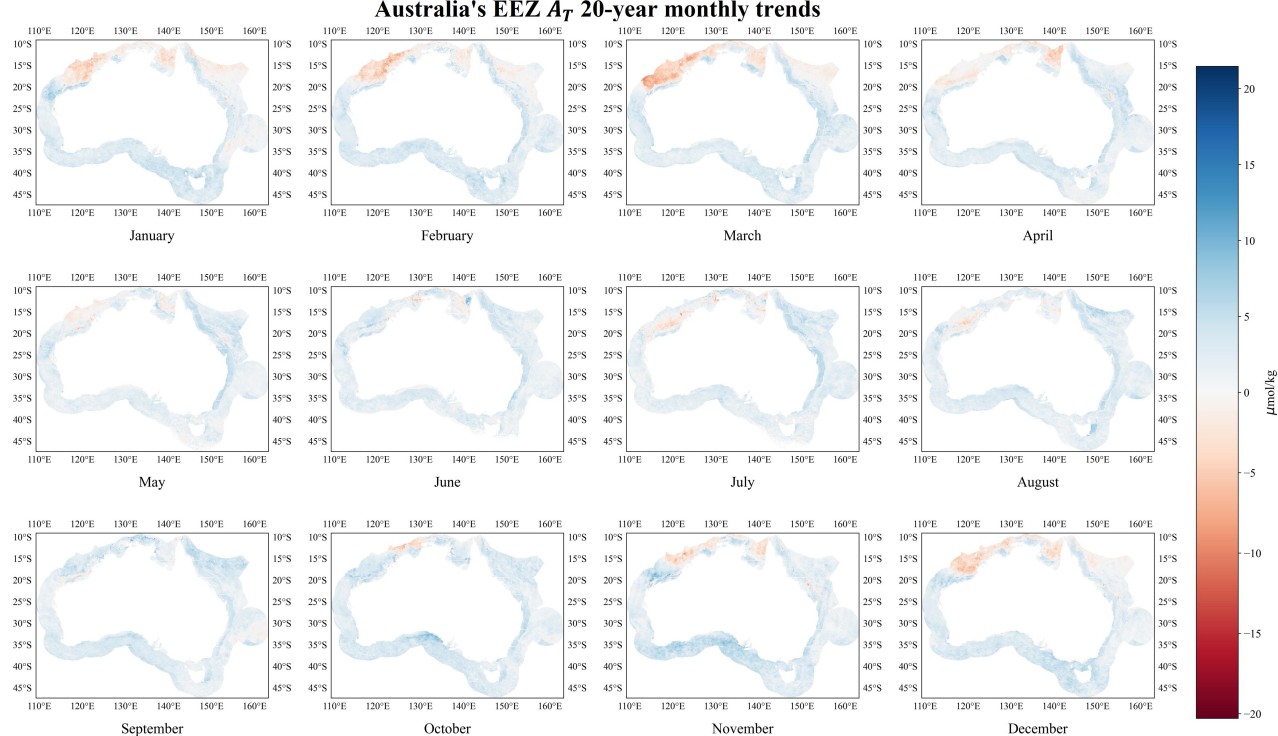

**Figure 4.** The monthly $A_T$ trends across the Australian EEZ over a span of 20 years.

along the southern edge of the continent, indicating an influx of higher $A_T$ waters or local biological processes affecting carbonate chemistry. April to June: There is a marked transition in $A_T$ concentrations with the onset of autumn and winter in the southern hemisphere. April shows relatively high concentrations, especially in the south-eastern part of the EEZ, which may be related to seasonal changes in water mass movement or biogeochemical cycling. July to September: The maps for these months show the highest $A_T$ values in the southernmost regions, suggesting winter conditions that may favour dissolution of carbonate minerals or reduced biological uptake of carbonate ions. October to December: As the Southern Hemisphere moves into winter, the $A_T$ of the EEZ increases: As the Southern Hemisphere moves into spring and summer, there is a gradual decrease in $A_T$ concentrations, particularly visible in the northern parts of the EEZ. The lighter colours may reflect changes in biological activity, such as increased photosynthesis leading to higher carbon dioxide uptake and hence lower $A_T$.

Figure 5 representation captures the longitudinal and latitudinal trends of total $C_T$ concentrations across the Australian EEZ over a span of twenty years. The trend of $C_T$ is characterised by a clear seasonal pattern. Due to increased metabolic activities and upwelling events that bring nutrient-rich deeper waters to the surface, the concentration of $C_T$ increases during the summer months (December-February) along the southern coast. As the seasons progress (March-May), there is a visible decrease in $C_T$ concentrations, especially evident in the northern regions, which could be attributed to the beginning of phytoplankton blooms consuming $CO_2$. The winter months (June-August) show a general reduction in $C_T$ levels across the EEZ, with the
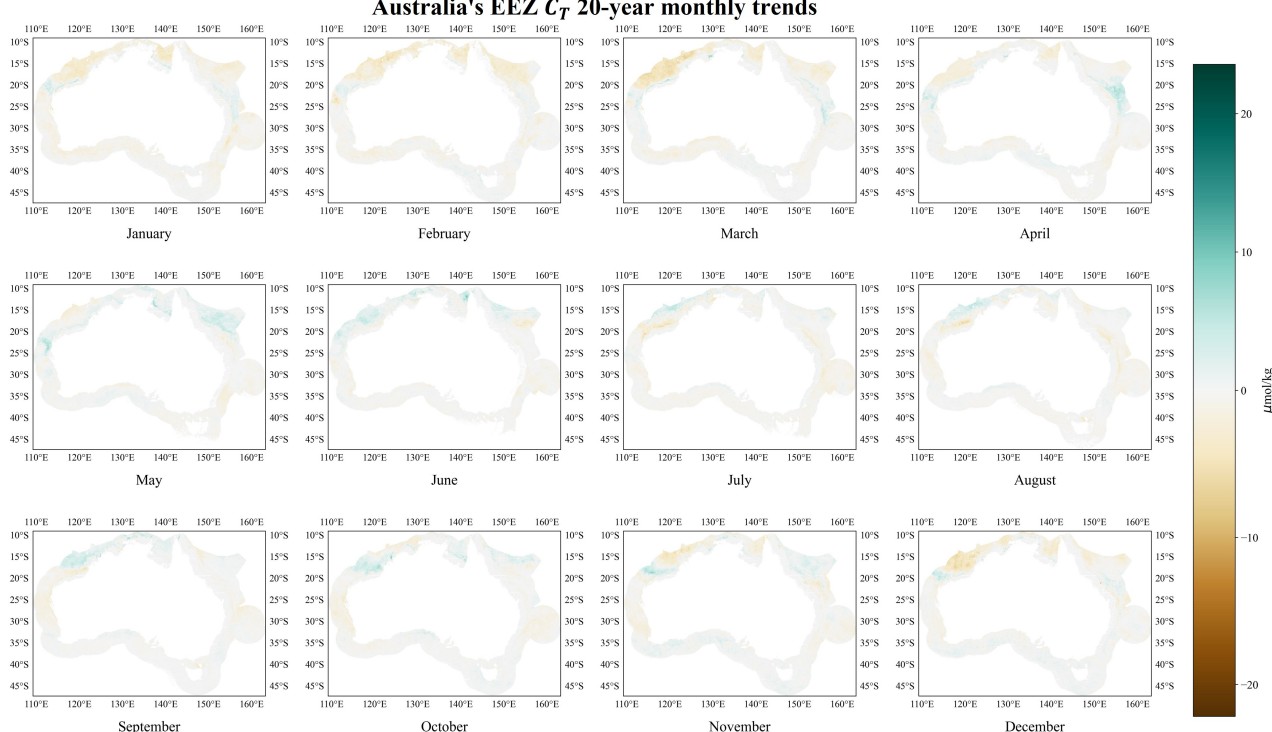

**Figure 5.** The monthly $C_T$ trends across the Australian EEZ over a span of 20 years.

lowest concentrations typically occurring around August. This trend reflect a decreased biological activity during the colder months. A resurgence in $C_T$ concentrations is observed in spring (September-November), especially along the northern and central parts, driven by the decay of organic matter as temperatures rise and sunlight increases. Further, the geographical variation of $C_T$ in the EEZ is also significant. Due to consistent tropical temperatures that stabilise metabolic rates and $CO_2$ solubility, generally shows less variation in $C_T$ concentrations throughout the year in the northern EEZ. In contrast, the South shows significant seasonal variations influenced by temperate climate dynamics, including variations in water temperature and biological productivity.

Figure 6 provides a detailed visual representation of pH trends across the Australian EEZ over a span of 20 years, delineated by month from January to December. The color gradient, ranging from blue to red, reflects pH changes, with blue indicating an increase in pH and red indicating a decrease. Notably, there is a distinct seasonal variation in pH levels, particularly evident around the southern edges of the EEZ, where shifts are more pronounced during certain months. For example, during the Southern Hemisphere's summer months (December through February), there is a notable decrease in pH, especially evident along the southern coast, indicative of upwelling events that bring more acidic, CO2-rich waters to the surface. Conversely, during winter and spring (June through September), there are areas, especially in the northern regions, where pH levels appear relatively stable or show slight increases. This could be linked to decreased biological activity and a reduction in the con-





**Australia's EEZ pH 20-year monthly trends**

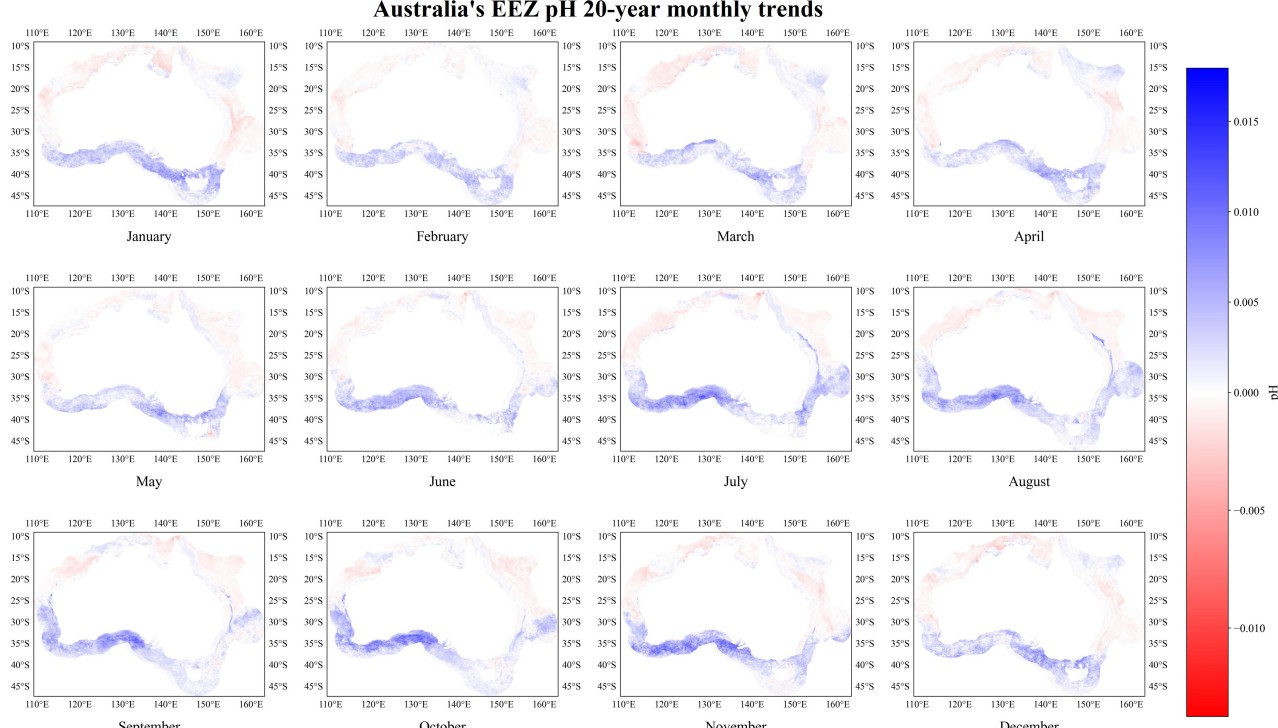

**Figure 6.** The monthly pH trends across the Australian EEZ over a span of 20 years.

sumption of carbon dioxide through photosynthesis. Spatially, the outer regions of the EEZ consistently show less variation in pH compared to the coastal areas. This stability might be attributed to the buffering capacity of the open ocean, which is less susceptible to rapid changes in acidity compared to coastal waters that are more influenced by terrestrial inputs, local upwelling events, and seasonal biological cycles.

Figure 7 comprehensively illustrates the spatial and temporal trends of $NO_3^-$ concentrations across EEZ over a period 230 of 20 years, presented in a monthly sequence from January to December. Throughout the year, $NO_3^-$ distributions exhibit significant seasonal variability. The highest concentrations, shown in deeper greens, are primarily observed during the Southern Hemisphere's autumn and winter months (March to August). These patterns suggest a strong seasonal influence, possibly driven by changes in water column mixing, upwelling events, and nutrient inputs from riverine sources. The winter months particularly show more extensive areas of high $NO_3^-$ concentrations extending further northwards, reflecting the influence of 235 cooler water movements and increased nutrient mixing from deeper layers.

Conversely, the summer months (December to February) display a reduction in $NO_3^-$ levels across most of the region, with concentrations typically retreating towards the southern and deeper waters. This seasonal reduction is likely due to increased stratification of the water column and higher biological uptake during warmer periods.



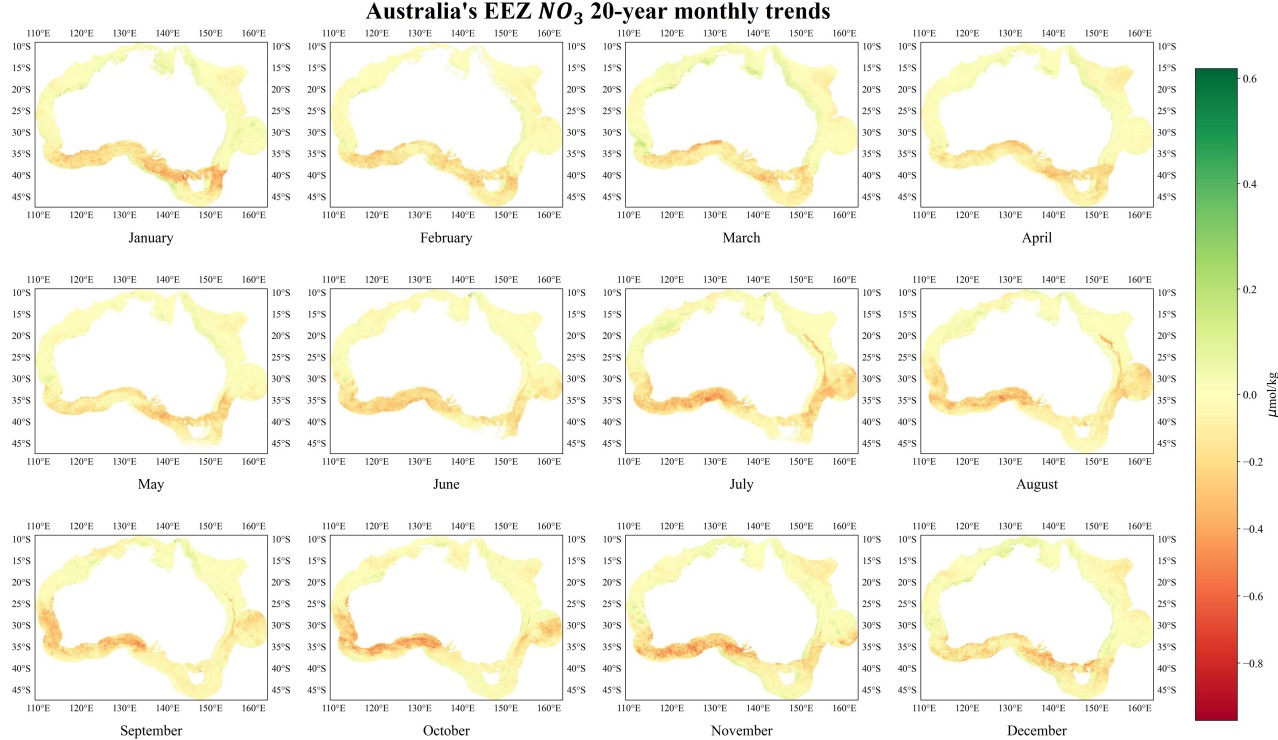

**Figure 7.** The monthly $NO_3^-$ trends across the Australian EEZ over a span of 20 years.

Spatially, the highest concentrations are consistently noted along the southern edges of the EEZ and diminish as one moves northwards, highlighting latitudinal gradients in nutrient availability. This pattern aligns with known oceanographic processes in the region, which influence nutrient dynamics and distribution, including the East Australian Current which tends to dilute nutrient concentrations as it flows poleward along the eastern coast.

Figure 8 represents monthly snapshots over a span of two decades, provides a detailed cartographic depiction of $PO_4^{3-}$ concentration trends across the Australian EEZ. Throughout the year, the observed trends reveal a distinct seasonal variation in $PO_43-$ levels, with higher concentrations typically manifesting during the cooler months (May to August). This pattern suggests a strong linkage with biogeochemical cycles and oceanic processes such as upwelling, which are known to be more active during these months due to prevailing wind patterns and water column stratification dynamics.

During the summer months, particularly from December through February, the visual data illustrates a clear decrease in $PO_43-$ levels across most regions within the EEZ. This seasonal decline is likely due to increased biological uptake by phytoplankton during periods of higher light availability and temperature, which promotes photosynthesis.

Figure 9 provides a systematic monthly depiction of $Si(OH)_4$ concentration trends within the Australian EEZ over a span of twenty years.



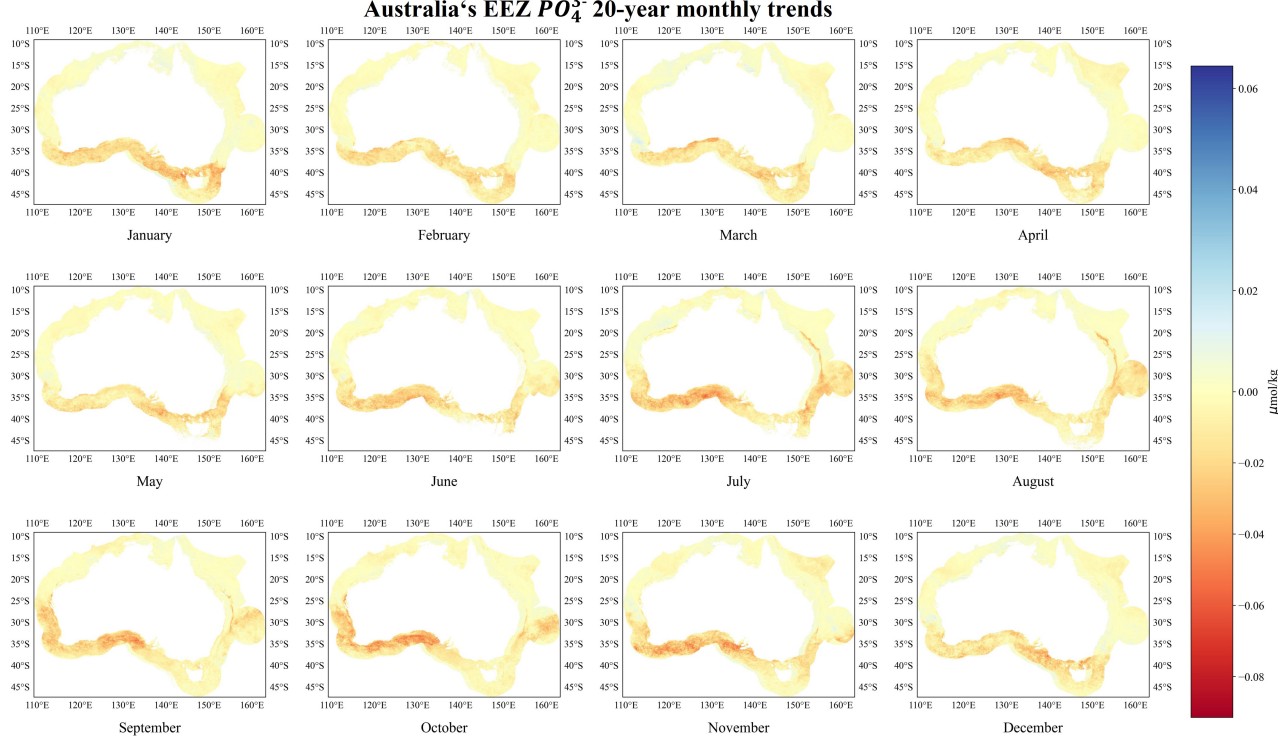

**Figure 8.** The monthly $PO_4^{3-}$ trends across the Australian EEZ over a span of 20 years.

The visual data indicate noticeable seasonal variability in $Si(OH)_4$ concentrations across the EEZ. During the winter and spring months (June to September), there is a pronounced increase in $Si(OH)_4$ levels, particularly in the southern regions, which are closer to upwelling zones that bring nutrient-rich waters from the deeper parts of the ocean to the surface. This seasonal peak is likely driven by the increased nutrient upwelling associated with the Southern Ocean's dynamics and the westerly winds. Conversely, during the summer months (December to February), there is a significant reduction in $SiOH_4$ concentrations, especially in the northern parts of the EEZ. This pattern may be influenced by higher biological uptake of $Si(OH)_4$ by diatoms and other siliceous phytoplankton, whose growth is enhanced by the warmer temperatures and increased sunlight.

Spatially, the highest concentrations of $Si(OH)_4$ are consistently observed in the deeper waters to the south, aligning with the regions of strong oceanic currents that transport colder, nutrient-rich waters northwards along the western and southern coasts. In contrast, the northern tropical and subtropical regions exhibit lower concentrations throughout the year, likely due to the different hydrographic conditions and lesser influence from the nutrient-rich Antarctic waters.

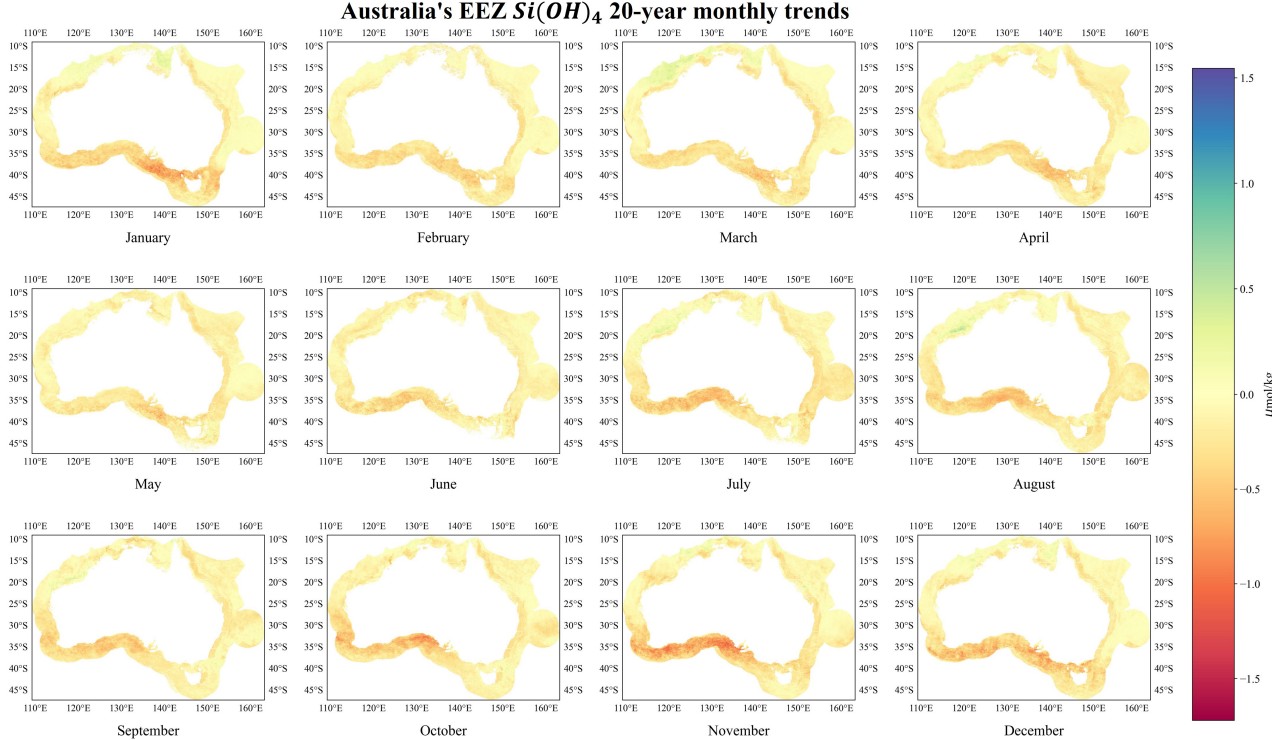

**Figure 9.** The monthly $Si(OH)_4$ trends across the Australian EEZ over a span of 20 years.

## 4.2 Accuracy Verification

### 4.2.1 5-fold Cross Validation

Figure 10 illustrates a comprehensive evaluation of model predictions versus measured values for several key oceanic biogeochemical parameters, graphically depicted through scatter plots. Each plot correlates the actual measured values on the x-axis with the predicted values on the y-axis, for the parameters $A_T$, $C_T$, pH, $NO_3^-$, $PO_4^{3-}$, and $Si(OH)_4$. The data points are color-coded according to depth, with shades ranging from light blue (shallow) to dark red (deep), providing a visual distinction of measurements taken at various ocean depths. RMSE and $R^2$ are prominently displayed on each plot, offering quantitative insights into the accuracy and reliability of the model predictions across different biogeochemical parameters. $A_T$ and $C_T$ plots show relatively high $R^2$ values, suggesting that the model predictions for $A_T$ and $C_T$ closely follow the observed data trends. However, the RMSE indicates that the absolute errors in predictions still present a challenge.

pH and $PO_4^{3-}$ display a tight clustering around the line of perfect agreement (dashed line), indicating strong model performance, especially for pH with an $R^2$ value of 0.80. $NO_3^-$ and $Si(OH)_4$ show considerable scatter, particularly at higher concentration ranges, suggesting variability in the model's predictive accuracy at elevated nutrient levels.




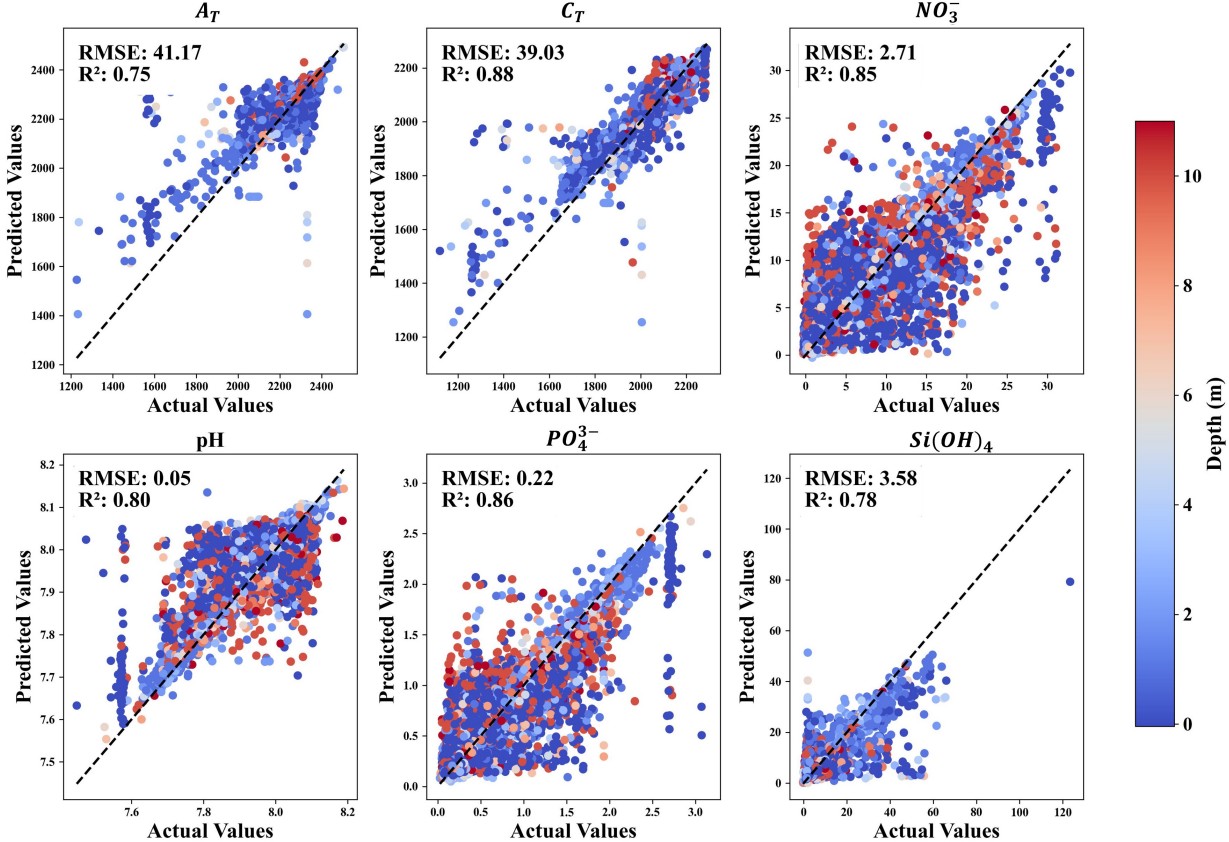

**Figure 10.** Validation of Random Forest Regression Modelling Results for Marine Biogeochemical Elements. The color of the dots represent the location of the water depths (water depth values are taken as absolute values).

The comprehensive visualization of feature importance across various models dedicated to predicting key oceanic biogeochemical parameters, as assessed by a machine learning framework presented in Figure 11. The depicted bar charts illustrate the relative importance of different features for predicting specific marine biogeochemical elements such as $A_T$, $C_T$, pH, $NO_3^-$, $PO_4^{3-}$, and $Si(OH)_4$. These features include both environmental parameters like temperature (Pres, SST) and biogeochemical indices (CHLA for chlorophyll a, POC for particulate organic carbon), alongside spectral data from multiple bands (MTORI Bands 1 through 9) of the MODIS Terra satellite. Each panel represents a distinct predictive model, with the x-axis listing the features and the y-axis quantifying their importance as determined by the model. The colors in each bar chart correspond to specific biogeochemical parameters, providing a clear visual representation of how each feature contributes to the prediction accuracy of each parameter. From the charts, it is evident that different features have varying degrees of influence on the models, depending on the specific biogeochemical parameter being predicted. For instance, MTORI Band 5 shows significant importance across several parameters, highlighting its utility in ocean chemistry modeling. Conversely, other bands and fea-

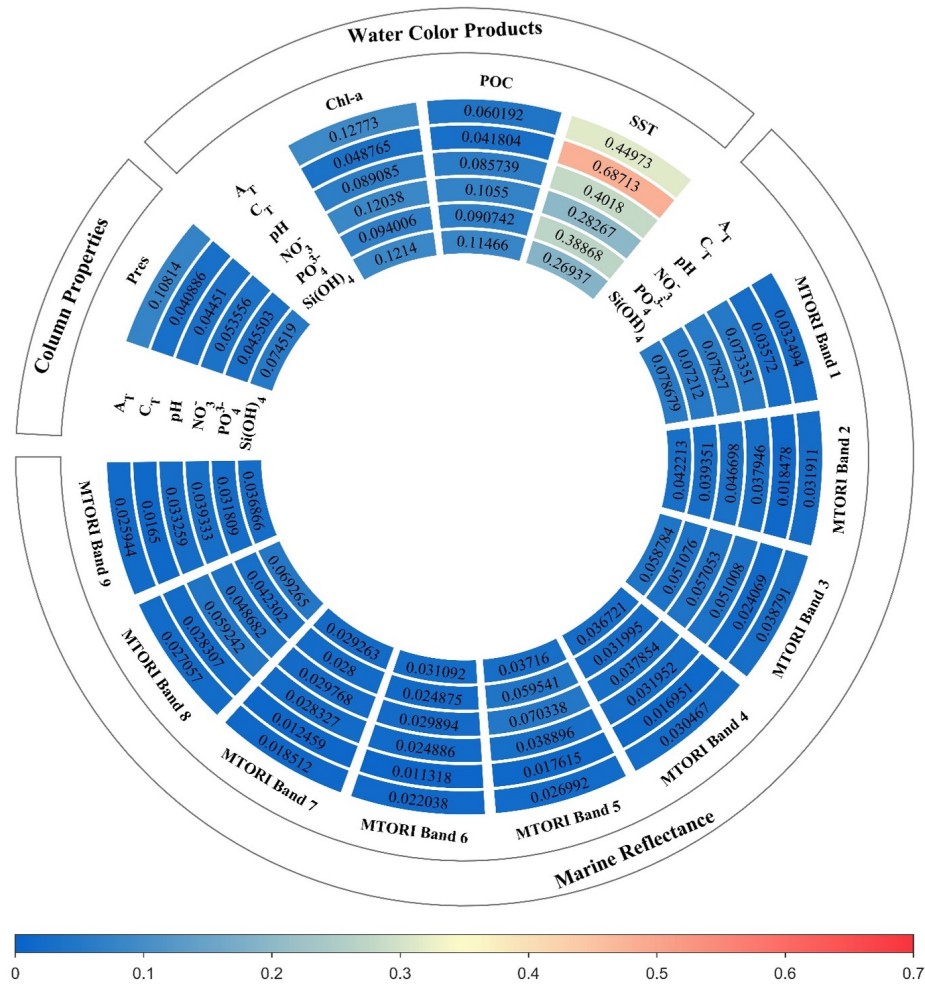

**Figure 11.** Importance of input features to the reconstruction model for each target element.

tures demonstrate more specialized relevance, indicating that their influence is more pronounced for certain biogeochemical elements than others.

### 4.2.2   Validation Against Independent Observations

The Australian Integrated Marine Observing System (IMOS) National Reference Stations (NRS) are a pivotal component in the nation's effort to monitor marine environments. The stations collect a diverse array of biogeochemical data (Yellow triangle in Figure 12), which is essential for understanding long-term changes in marine ecosystems due to environmental pressures like climate change and human activities. Located strategically around Australia's coast, each station provides critical data on parameters such as salinity, temperature, dissolved oxygen, and nutrients through continuous and systematic observations. The reconstructed parameters of $A_T$, $C_T$, pH, $NO_3^-$, $PO_4^{3-}$ and $Si(OH)_4$ were rigorously validated using independent ob-



**Figure 12.** Distribution of independent observation data and results of independent validation for each marine biogeochemical product. Satellite image obtained from Bing Maps © Microsoft.



servational datasets derived from Argo floats and extensive multi-year observations at NRS. These validation datasets, which were deliberately excluded from the regression model construction phase to preserve the integrity of the validation process, are
300 distributed across the entire Australian EEZ, as shown in Figure 12. Comparison of these independently observed data with the reconstruction results showed a high degree of agreement, confirming the robustness of the reconstruction methodology. In particular, multi-year in situ pH data from Argo floats and nutrient salinity time series from NRS were instrumental in confirming the accuracy of the reconstructed biogeochemical and physical parameters of the marine environment.

Figure 12 shows in detail the time-series comparison of six elemental reconstructions based on observational data. This
verification includes six components, labelled (a) to (f), each demonstrating the validation of reconstructed ocean parameters against an ensemble of independent observations collected over time.

In Figure 12 (a), a discernible stable trend in $A_T$ is observable, with values predominantly aggregating around 2300 $\mu$mol/kg. The scatter plot elucidates a tight clustering around the predicted values, epitomizing high accuracy in the reconstruction. This is quantitatively supported by an $R^2$ value of 0.74 and an RMSE of 15.84, indicative of robust model performance. Figure
12(b) delineates the behavior of $C_T$, which similarly exhibits a stable pattern, with measurements consistently approximating 2100 $\mu$mol/kg. The dispersion of data points is marginally broader than that observed for $A_T$, mirrored by an $R^2$ of 0.67 and a heightened RMSE of 21.31, suggesting a modest decrement in model accuracy for $C_T$ relative to $A_T$. Proceeding to Figure 12(c), the reconstructed pH values display a variance yet predominantly confine within a narrow interquartile range of 7.9 to 8.1. The alignment between the predicted and observed values is compelling, denoted by an $R^2$ of 0.72 and an RMSE of 0.07,
which underscores the precision of the pH reconstructions.

Figures 12 (d), (e), and (f) scrutinize the elements of $NO_3^-$, $PO_4^{3-}$, and $Si(OH)_4$ respectively. Notably, $NO_3^-$ concentrations manifest a broad spectrum of values with a relatively scattered distribution. Despite these variances, the correlation remains robust with an $R^2$ of 0.77. However, a relatively elevated RMSE of 0.161 could signify sporadic discrepancies between predicted and observed values. Conversely, $PO_4^{3-}$ concentrations, typically ranging between 0.5 and 1.5 $\mu$mol/kg, exhibit a
320 dense clustering of points, indicative of consistent and accurate predictions—corroborated by an $R^2$ of 0.79 and an RMSE of 0.12. $Si(OH)_4$, manifesting the most extensive range of values among the panels from approximately 20 to 120 $\mu$mol/kg, shows a more pronounced scatter, reflected in a lower $R^2$ of 0.72 and a higher RMSE of 1.75 compared to the other nutrients.

Overall, the trends in the image suggest that the reconstructed parameters generally align well with the observational data, albeit with varying degrees of accuracy across different parameters. This demonstrates the effectiveness of the reconstruction
models in replicating true marine conditions, thus providing a reliable foundation for studying marine biogeochemistry and environmental impacts on marine ecosystems.

### 4.2.3 Uncertainties in Satellite Remote Sensing and Field Observations

Satellite remote sensing serves as a crucial tool for acquiring extensive, long-term datasets of ocean reflectance, essential for monitoring marine environments. However, the inversion accuracy of satellite data is often compromised by natural phenomena
and sensor limitations. Weather conditions such as clouds and fog can obscure the observation areas, leading to compromised data quality or complete data loss, presenting significant challenges to satellite data retrieval.



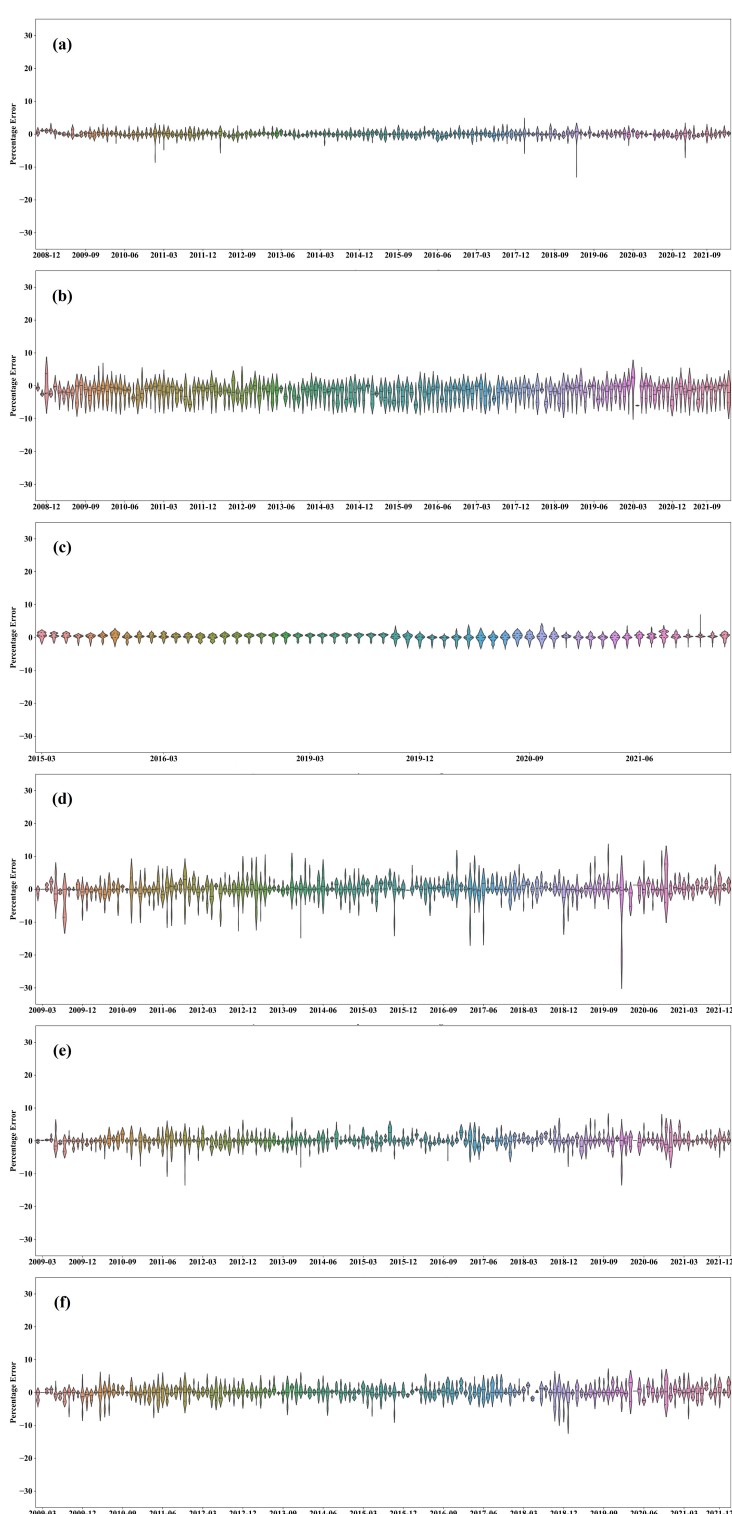

**Figure 13.** Distribution of independent observation data and results of independent validation for each marine biogeochemical product.





To mitigate the effects of cloud cover and shadows, this study utilizes the MODIS satellite's high temporal resolution alongside the robust data processing capabilities of the Google Earth Engine (GEE) platform. By employing the $median()$ function on GEE, we generate optimal monthly composite images that reduce cloud-related disturbances, although discrepancies due to varying illumination conditions persist. Furthermore, the scarcity of consistent, continuous in-situ measurements, coupled with the heterogeneity in data formats, measurement protocols, and the clarity of data provenance, constrains the utility of field data in analyzing long-term, large-scale trends in nutrient concentrations.

Figure 13 provides a comprehensive visualisation of the error distribution for six reconstructed ocean parameters - $A_T$, $C_T$, pH, $NO_3^-$, $PO_4^{3-}$ and $Si(OH)_4$ - analysed over a longer time scale from 2008 to 2021. Figure 13 (a) to (f), represents the monthly product percent error distribution for each parameter, indicating the deviation of the reconstructed values from independent observational datasets. Figure 13 (a) $A_T$ and Figure 13 (b) $C_T$ demonstrate comparable error distributions, exhibiting minor periodic fluctuations and maintaining an error within $\pm10\%$. This indicates a relatively stable and accurate model performance across the time series. Figure 13 (c) presents the pH errors, which exhibit a distinct pattern compared to other nutrients. The tight clustering of errors around zero interspersed with periodic, systematic shifts in the pH reconstruction error distribution could suggest the influence of specific calibration intervals of the model or discernible environmental perturbations. These deviations from baseline accuracy in the pH measurements reflect dynamic oceanic conditions or adjustments in the methodological approachs. Figure 13 (d) $NO_3^-$ and Figure 13 (e) $PO_4^{3-}$ display a greater degree of variation in the error distribution, with broader ranges, particularly in $NO_3^-$, where the variability is more pronounced. This could indicate differing levels of sensitivity in the reconstruction algorithms to environmental factors or variations in data quality and availability. Figure 13 (f) for $Si(OH)_4$, the error distribution is similar to that of $NO_3^-$, indicating the difficulties in reconstructing $Si(OH)_4$ concentrations, which may be influenced by complex biogeochemical interactions that are less prevalent in parameters such as $A_T$ and $C_T$.

The error distribution is crucial for understanding the uncertainty in satellite-derived oceanic parameter estimates. This emphasises the need for continuous refinement of algorithms and the incorporation of diverse data sources to enhance the accuracy and reliability of these vital biogeochemical estimations. The discrepancies in uncertainty across the parameters illustrate the intricacy of marine chemistry inversion and identify areas where further attention to algorithm enhancement and calibration could result in substantial gains in the accuracy of satellite-derived marine data.

In addressing these challenges, our methodology ensures consistency by sourcing data from identical institutions or sensors, minimizing variability introduced by differing data collection techniques. Despite these precautions, the integration of multiple data sources introduces inherent uncertainties, as reliance on a single sensor does not satisfy the input requirements for the CANYON-B model. This approach assumes a uniform distribution of input observations across the study area, acknowledging the potential for inconsistencies in data integration but striving for the broadest possible representation of environmental conditions within the analysis framework.

## 5 Conclusions

In the context of oceanographic research, leveraging satellite remote sensing technology is indispensable for assessing large-scale and long-term variations in ocean nutrient salts. Utilizing MODIS ocean reflectance imagery, coupled with advanced algorithmic approaches such as CANYON-B and Random Forest modelling, has demonstrated substantial potential in inverting nutrient concentrations with enhanced precision. Utilizing MODIS ocean reflectance imagery, coupled with advanced algorithmic approaches such as CANYON-B and Random Forest modelling, has demonstrated substantial potential in inverting
nutrient concentrations with enhanced precision.

Our results indicate that while the Random Forest and CANYON-B models are adept at processing the vast datasets provided by MODIS, they also require meticulous calibration and validation against in situ data to ensure accuracy. This calibration was performed using a combination of direct measurements and intercomparisons with established satellite products, enhancing the credibility of the nutrient salt reconstructions. Through this rigorous approach, our study contributes to the ongoing efforts
to refine satellite data inversions, providing a more reliable foundation for the scientific community to study oceanic nutrient dynamics and their implications on marine ecosystems. The study underscores the potential of integrating machine learning techniques with traditional remote sensing methods to enhance the predictive accuracy of ocean nutrient concentrations, ultimately supporting global marine environmental monitoring and management.

*Data availability.*   The datasets generated is available on Science Data Bank (https://doi.org/10.57760/sciencedb.09331) (Zhang et al., 2024).

*Author contributions.*   Lizhe Wang provided scientific ideas, reviewed the paper and contributed to the revising of figures and words of this paper; Xiaohan Zhang collected the datasets, wrote the codes, analyzed the data, plotted the figures and wrote the paper; Jining Yan and Sheng Wang contributed to the revising of figures and words of this paper.

*Competing interests.*   The contact author has declared that none of the authors has any competing interests.

*Acknowledgements.*   This work was jointly supported by the National Natural Science Foundation of China (41925007 and U21A2013) .



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
