# Peer review of "High-spatiotemporal reconstruction of biogeochemical dynamics in Australia integrating satellites products and in-situ observations (2000–2022)"

_Earth System Science Data, 2024_

## Author Comment (AC2)

**Response to reviewers**

**List of Main Actions (LMAs):**

LMA1: Added a clear explanation of the meaning of Ti(x) in the revised manuscript.

LMA2: Three citations were added to the literature on Random Forest regression algorithms and three citations to studies relevant to the research context.

LMA3: Responded to all of the reviewer's comments and questions regarding this paper in a response letter.

**Response to Associate Editor:**

Dear xiaohan zhang,

Thank you very much for your response to the interactive comments on your preprint under peer review for Earth System Science Data:

Title: High-spatiotemporal reconstruction of biogeochemical dynamics in Australia integrating satellites products and in-situ observations (2000–2022)

Author(s): Xiaohan Zhang et al.
MS No.: essd-2024-219
MS type: Data description paper

We noticed that you have already posted a sufficient number of author comments (ACs). The final response phase could now be completed to continue with the review process.

Please complete the final response phase through the button "Finalize" at: https://editor.copernicus.org/essd-2024-219/final-response

Please log in using your Copernicus Office user ID 761271 at your earliest convenience but no later than 06 Feb 2025.

You are invited to monitor the processing of your manuscript via your MS overview at: https://editor.copernicus.org/ESSD/my_manuscript_overview

Thank you very much in advance for your cooperation. In case any questions arise, please do not hesitate to contact me.

Kind regards,

The editorial support team
Copernicus Publications
editorial@copernicus.org

**Response:**

Thank you very much for giving us the opportunity to submit the revised manuscript. We have carefully revised the manuscript according the comments and suggestions. The problems pointed by the reviewer have been fixed in the revised manuscript. Responses to the reviewers are given as below.

**Reviewer comments:**

This work presents a new framework combining the CANYON-B model and the Random Forest Regression approach for reconstructing marine biogeochemical dynamics in Australia's Exclusive Economic Zone (EEZ) at high spatial (1 km) and temporal resolutions. This study fills a gap in traditional interpolation methods for the analysis of dynamic ocean processes at large scales, especially in complex gradient and highly dynamic regions. In addition, the article makes full use of MODIS Terra products with multi-source in situ observations from Argo, gliders and ANMN to achieve accurate modeling across temporal and spatial scales. The innovative approach of combining MODIS satellite remote sensing data with in situ observations provides an important reference for future ocean monitoring. The manuscript is clearly structured, easy to read and well organized. It is recommended for acceptance with modifications.

Here are my review comments:

1. Based on the manuscript's description, it appears that the authors produced a sea surface biogeochemistry data product. It is recommended that the authors explicitly and clearly highlight this in the article to ensure readers fully understand the nature and scope of the data product.

2. The manuscript describes using Argo buoy data as input to the CANYON-B model, followed by validation using the same Argo data. Why not directly use the biogeochemical parameters from Argo for the reconstruction?

3. In Table 1, the use of the separator "-" in the "Contains Elements" column is potentially confusing, particularly with entries like "Chl-a." It is suggested to replace the "-" with a comma (",") to improve clarity.

4. Line 145: In Equation 1, it is recommended that the meaning of "Ti(x)" be clearly explained.

5. Line 385: Some of the references are old (e.g., from 1994 and 2004), and it is suggested to add the latest research support on random forests in recent years.

6. Why not use IMOS real data directly to train the random forest model?

7. How to control the date correspondence between point data and MODIS impacts.

8. The number of citations in the manuscript appears to be relatively low. It is suggested to add more relevant and recent references to provide broader scientific context and support the study's methodology and findings.

**Response to reviewers' comments:**

**Overall review comments:** This work presents a new framework combining the CANYON-B model and the Random Forest Regression approach for reconstructing marine biogeochemical dynamics in Australia's Exclusive Economic Zone (EEZ) at high spatial (1 km) and temporal resolutions. This study fills a gap in traditional interpolation methods for the analysis of dynamic ocean processes at large scales, especially in complex gradient and highly dynamic regions. In addition, the article makes full use of MODIS Terra products with multi-source in situ observations from Argo, gliders and ANMN to achieve accurate modeling across temporal and spatial scales. The innovative approach of combining MODIS satellite remote sensing data with in situ observations provides an important reference for future ocean monitoring. The manuscript is clearly structured, easy to read and well organized. It is recommended for acceptance with modifications.

**Author's Response:**

We sincerely appreciate the reviewer's positive feedback and their recognition of the significance of our work. We are grateful for the acknowledgment of our novel framework, which integrates the CANYON-B model with Random Forest Regression to reconstruct marine biogeochemical dynamics in Australia's Exclusive Economic Zone (EEZ) at high spatial and temporal resolutions.

We particularly appreciate the reviewer's recognition of our effort to bridge gaps in traditional interpolation methods, especially in highly dynamic and complex gradient regions. The integration of MODIS Terra satellite products with multi-source in situ observations from Argo, gliders, and ANMN indeed enhances the accuracy and robustness of our model across different scales. We are pleased that the reviewer considers our approach to provide an important reference for future ocean monitoring.

In response to the reviewer's recommendation for modifications, we have carefully revised the manuscript to further improve clarity, structure, and scientific rigor. The specific modifications are detailed in our response to the comments. We hope that the revised manuscript meets the reviewer's expectations and look forward to their further insights.

Once again, we greatly appreciate the reviewer's constructive comments and their recommendation for acceptance.

**Review comments 1:** Based on the manuscript's description, it appears that the authors produced a sea surface biogeochemistry data product. It is recommended that the authors explicitly and clearly highlight this in the article to ensure readers fully understand the nature and scope of the data product.

**Author's Response:**

Thanks for your suggestion. The product produced by our work is indeed a "marine surface data product", as we explain in the abstract of the revised manuscript.

Line 9:

"……. This work reconstructs the concentrations of these **marine surface** biogeochemicals at the sea surface within Australia's Exclusive Economic Zone over the period from 2000 to 2022 on a 1-kilometre scale. ……."

**Review comments 2:** The manuscript describes using Argo buoy data as input to the CANYON-B model, followed by validation using the same Argo data. Why not directly use the biogeochemical parameters from Argo for the reconstruction?

**Author's Response:**

We made this choice for two primary reasons. First, only a small subset of Argo floats is equipped to monitor biogeochemical parameters, and their spatial and temporal distribution is highly heterogeneous within our study area. Directly incorporating these data into the model would introduce bias in model fitting and reduce the accuracy of the reconstructed biogeochemical products.

Second, the biogeochemical parameters measured by Argo floats serve as an independent dataset to assess the performance of the CANYON-B model. By using Argo-derived pressure, temperature, salinity, and oxygen concentration as inputs, we can evaluate the model's ability to estimate biogeochemical parameters accurately. This step ensures a rigorous validation process and confirms that Argo data can be reliably used as an input for CANYON-B.

**Review comments 3:** In Table 1, the use of the separator "-" in the "Contains Elements" column is potentially confusing, particularly with entries like "Chl-a." It is suggested to replace the "-" with a comma (",") to improve clarity.

**Author's Response:**

Thanks for your suggestions, which have been revised in the latest manuscript.

**Table 1.** Multi-source input datasets

| Dataset Name | Data Type | Contains Elements | Timespan | Source |
|---|---|---|---|---|
| MODIS Terra Marine Reflectance Products | Raster | 1km ocean reflectance data from Terra MODIS bands 8-16 | 2000–2022 | GEE |
| MODIS Terra Ocean Water Color Products | Raster | Chl-a, POC, SST(Resampling from 4km to 1km spatial resolution) | 2000–2022 | GEE |
| Australian Glider Observations | Point | Chl-a, Current, Optical properties, Oxygen, Salinity, Temperature, Water pressure | 2008–2024 | AODN |
| ANMN Monitor Data | Point | Chl-a, Oxygen, Salinity, Temperature, Turbidity, Water pressure | 2007–2024 | AODN |
| Argo Buoy Data | Point | Salinity, Temperature, Water pressure | 1999–2024 | AODN |

**Review comments 4:** Line 145: In Equation 1, it is recommended that the meaning of "Ti(x)" be clearly explained.

**Author's Response:**

We provide a clear explanation of the meaning of Ti(x) in the revised manuscript.

Line 155-156:

"The predictor variables Ti(x) used in the Random Forest regression include column properties (Pres), water color products (Chl-a, POC, and SST), and MODIS Terra satellite-derived ocean reflectance products from bands 1–8."

**Review comments 5:** Line 385, Some of the references are old (e.g., from 1994 and 2004), and it is suggested to add the latest research support on random forests in recent years.

**Author's Response:**

Thanks to your suggestion, the latest manuscript has been added with 2 newest studies having to do with random forest regression.

Line 146:⋯⋯decision trees and combining their predictions(Wang et al., 2021).

Line 154-155:⋯⋯improves the accuracy and stability of the predictions(Grinsztajn et al., 2022; Guillon et al., 2024).

**Review comments 6:** Why not use IMOS real data directly to train the random forest model?.

**Author's Response:**

While the IMOS real data is evenly distributed over the study period, there are two main reasons why it was not used as the sole source of training data. First, IMOS provides biogeochemical concentration data only after 2008, whereas our reconstructed product spans the period from 2000 to 2022. Given that Random Forest is a data-driven approach, training the model exclusively on post-2008 data and extrapolating to pre-2008 conditions would be unreliable.

Second, the spatial distribution of IMOS observations is uneven, with sparse coverage in the northern part of the study area. Relying solely on IMOS data for training could introduce spatial biases and negatively impact the accuracy of the estimated biogeochemical concentrations.

**Review comments 7:** How to control the date correspondence between point data and MODIS impacts.

**Author's Response:**

When extracting the pixel values from MODIS images corresponding to in situ biogeochemical concentration measurements, we ensured that the image acquisition date matched the observation date. If no MODIS image was available for the exact date, we used data from the nearest available day, either the preceding or following day, as a substitute.

For example, temperature (T), salinity (S), and oxygen ($O_2$) concentrations measured on July 10, 2010, were used as inputs to the CANYON-B model to estimate biogeochemical concentrations. These estimates were then matched with pixel values from MODIS image products acquired on the same day at corresponding locations. If no image was available on July 10, pixel values from July 9 or July 11 were used instead.

Once a sufficient number of correspondences were established and evenly distributed, Random Forest regression was applied to learn the relationship between biogeochemical concentrations and MODIS pixel values. The trained model was then generalized to other times and locations to reconstruct biogeochemical concentrations across the entire study area.

**Review comments 8:** The number of citations in the manuscript appears to be relatively low. It is suggested to add more relevant and recent references to provide broader scientific context and support the study's methodology and findings.

**Author's Response:**

Thanks to your suggestion, citations to three recent studies on marine biochemical parameters have been added to the latest manuscript.

Line 26-27:

⋯⋯protecting marine life(Fourrier et al., 2022).

Line 43:

⋯⋯vary significantly over short distances and time periods(Mignot et al., 2023).

Line 49:

⋯⋯high costs associated with data collection(Asselot et al., 2024).